# Calcium Signalling in Neurological Disorders, with Insights from Miniature Fluorescence Microscopy

**DOI:** 10.3390/cells14010004

**Published:** 2024-12-25

**Authors:** Dechuan Sun, Mona Amiri, Qi Meng, Ranjith R. Unnithan, Chris French

**Affiliations:** 1Neural Dynamics Laboratory, Department of Medicine, The University of Melbourne, Melbourne, VIC 3052, Australia; maami@student.unimelb.edu.au (M.A.); qmme@student.unimelb.edu.au (Q.M.); 2Department of Electrical and Electronic Engineering, The University of Melbourne, Melbourne, VIC 3052, Australia; r.ranjith@unimelb.edu.au

**Keywords:** calcium signalling, amyotrophic lateral sclerosis, Alzheimer’s disease, Parkinson’s disease, Huntington’s disease, schizophrenia, miniscope

## Abstract

Neurological disorders (NDs), such as amyotrophic lateral sclerosis (ALS), Alzheimer’s disease (AD), Parkinson’s disease (PD), Huntington’s disease (HD), and schizophrenia, represent a complex and multifaceted health challenge that affects millions of people around the world. Growing evidence suggests that disrupted neuronal calcium signalling contributes to the pathophysiology of NDs. Additionally, calcium functions as a ubiquitous second messenger involved in diverse cellular processes, from synaptic activity to intercellular communication, making it a potential therapeutic target. Recently, the development of the miniature fluorescence microscope (miniscope) enabled simultaneous recording of the spatiotemporal calcium activity from large neuronal ensembles in unrestrained animals, providing a novel method for studying NDs. In this review, we discuss the abnormalities observed in calcium signalling and its potential as a therapeutic target for NDs. Additionally, we highlight recent studies that utilise miniscope technology to investigate the alterations in calcium dynamics associated with NDs.

## 1. Introduction

Neurological disorders (NDs) comprise a range of pathological conditions that arise from the progressive and irreversible dysfunction of neurons in the central nervous system [1,2]. These disorders can be broadly classified into two main categories: neurology and psychiatry [3]. Conditions such as amyotrophic lateral sclerosis (ALS), Alzheimer’s disease (AD), Parkinson’s disease (PD), and Huntington’s disease (HD) all fall under the category of neurology, whereas schizophrenia is classified as a psychiatric disorder. Despite substantial funding being invested in the study of NDs, the underlying mechanisms behind these disorders are still not clear. The diagnosis of NDs primarily depends on clinicopathological features, which include both distinctive clinical symptoms and often the presence of characteristic brain lesions [4]. Furthermore, all approved medications for these disorders can only provide symptomatic relief, with limited long-term benefits [5,6,7]. Previous studies in clinical and animal models have identified several key factors contributing to the development and progression of NDs, including gene mutations, abnormal protein accumulation, and metabolic disruptions [8,9,10]. Notably, many of these factors have been shown to influence calcium signalling, which disrupts neuronal function and contributes to the progression of NDs [11,12,13].

Calcium is a ubiquitous second messenger that participates in a wide range of cellular processes in the central nervous system. It is crucial in regulating gene expression, synaptic transmission and plasticity, and multiple metabolism processes [14]. Dysfunction in calcium signalling can lead to either excessive or insufficient calcium levels within neurons [11]. Over time, these disruptions can trigger signalling for apoptosis and contribute to the development of NDs [15]. The delicate kinetic balance between calcium influx and efflux is critical for normal cellular functions [16]. Calcium influx is regulated by different types of channels and receptors. Among these, voltage-gated calcium channels and ligand-gated calcium channels are the most extensively studied [17,18]. These channels facilitate calcium entry into the cell in response to changes in membrane potential or the binding of specific ligands, respectively. Additionally, transient receptor potential channels become permeable to calcium ions when activated by external stimuli. Similarly, calcium entry can be induced by ligand-gated calcium channels, such as NMDA receptors (see Capiod, 2011 for review) [19]. On the other hand, calcium efflux is regulated through both direct and indirect pathways [20]. Utilising energy from ATP hydrolysis or the sodium gradient, ATP dependent pumps and sodium–calcium exchangers transport calcium out of the cells directly [21,22]. In contrast, indirect pathways temporarily store or buffer excess intracellular calcium to help prevent calcium overload [23,24,25]. Sarcoendoplasmic reticulum calcium ATPase pumps help maintain appropriate calcium levels in the cytosol by transporting calcium back into the endoplasmic reticulum (ER) for storage [23]. Similarly, mitochondrial calcium uniporters facilitate rapid calcium accumulation across the inner mitochondrial membrane, and calcium binding proteins bind to calcium ions to buffer intracellular calcium levels [24,25]. As calcium signalling is involved in numerous molecular processes, this pathway is reinforced as a promising target for ND treatments.

Since its invention, two-photon microscopy has become the most widely used method for studying calcium dynamics in NDs within in vivo settings [26]. This technique enables volumetric imaging of the brain and allows researchers to observe and investigate calcium transients with subcellular resolution. In recent years, a miniature fluorescence microscope (miniscope) utilising one-photon excitation has emerged as a valuable tool in the field [27]. This device enables the real-time recording of calcium activity in large neuronal populations within behaving animal models of NDs. Extensive evidence suggests that many of the neurological deficits observed in NDs may be attributed to functional impairments in calcium signalling within the neural network rather than solely to neuronal loss (see Section 3: Calcium Signalling in Neurological Disorders). These results indicate that therapies aimed at reconstructing and restoring neural circuit function may be more beneficial than those that focus solely on the characteristics of individual neurons. In this review, we discuss the abnormalities observed in intracellular calcium signalling and its potential as a therapeutic target for NDs, including AD, HD, PD, ALS, and schizophrenia. Furthermore, we provide a critical review of studies on NDs that investigate alterations in calcium dynamics across large neuronal populations using the miniscope.

## 2. Fluorescence Two-Photon Microscopy and Miniature One-Photon Microscopy

In 1990, Denk et al. invented the first two-photon laser scanning fluorescence microscope that could achieve subcellular resolution in biological tissues labelled with fluorescent dyes [28]. This breakthrough has significantly transformed neuroscience research, enabling volumetric brain imaging in live and even behaving animals [29]. At the same time, progress in the development of chemical calcium indicators has allowed for more detailed observation of neuronal calcium activity [30]. These indicators function by binding to calcium ions through chelation, which results in a change in their optical properties. Two commonly used classes of chemical calcium indicators are the Fura and Fluo families. Fura family indicators are excited by light at two different wavelengths, and the calcium concentration is determined by measuring the excitation ratio. In contrast, Fluo family indicators are based on single-wavelength excitation. Although this property makes them less accurate than their Fura counterparts, they offer certain advantages in terms of ease of use and simplicity in experimental setups. Since 2000, a new class of calcium indicators, genetically encoded calcium indicators, have begun to take shape and undergone substantial improvements, especially in terms of their response time, sensitivity, noise ratio, and reduced phototoxicity [31]. Moreover, these indicators can be targeted not only to specific cell populations but also to specific cell organelles with the proper calcium affinity. This targeting capability provides researchers with a unique tool for studying the calcium dynamics of specific neuronal populations when used in conjunction with two-photon microscopy. In addition to imaging techniques, several alternative methods are available for measuring the electrical signals associated with neuronal calcium activity. For example, patch clamp electrophysiology uses a glass pipette to measure calcium flux across neuronal membranes at the molecular level [32]. Calcium probes, such as calcium ion-selective microelectrodes or calcium-selective field-effect transistors, utilise calcium-selective membranes that specifically bind to calcium ions to monitor calcium ion concentrations and activities [33,34]. When exposed to a solution containing calcium ions, this selective binding generates a charge imbalance across the membrane, resulting in an electrical potential. Compared to calcium imaging methods, these techniques offer superior temporal resolution. However, they are not ideal for recording large neuronal populations and are primarily used in in vitro experiments due to the challenges associated with miniaturising the recording device. Additionally, although brain tissue can maintain a relatively stable pH and temperature, which does not severely affect the recording accuracy, interference from other ions and membrane degradation over time may reduce its sensitivity and cause signal drift, leading to inaccurate measurements. One major limitation of these techniques is their non-specificity. Calcium probes do not possess intrinsic capabilities for targeting specific neurons based on molecular markers. As a result, researchers often rely on electrophysiological signatures (e.g., firing frequency or firing patterns) to identify and differentiate between neuronal subtypes during experiments.

The two-photon microscope is typically built on a benchtop microscope, featuring an integrated mode-locked laser source, a laser scanning system, and a detector. However, the high cost of the laser source limits its widespread use. In vivo two-photon microscopy performed on live animals involves an essential step of creating an intracranial window in the region of interest, which may be located either superficially or deep within the brain. The superficial window is achieved by placing a thin cover glass on the brain’s surface, whereas the deep window requires the removal of overlying brain tissue and the insertion of a relay lens to access deeper brain regions. Then, the cortical activity can be captured through the window. However, the requirement of head fixation during experiments restricts its application to specific behavioural paradigms and can cause stress, affecting natural neural activity due to the lack of vestibular input. Additionally, the trade-off between the sampling frequency and spatial resolution limits the device to operate at a relatively low frequency of approximately 5–30 Hz. This may not be sufficient for capturing rapid changes in neuronal calcium dynamics across a large field of view.

Recently, the invention of one-photon miniaturised fluorescence microscopes (miniscopes) has made it possible to observe the spatiotemporal calcium activity at cellular resolution across large neuronal populations [35,36,37,38]. The working principle of the miniscope is very similar to conventional benchtop fluorescence microscopes. The system consists of several key components, including an illumination LED integrated with a half-ball lens, a filter set comprising emission and excitation filters in addition to a dichroic mirror, as well as an achromatic lens to enable light focusing. These components work together to direct stimulation light into the brain tissue and guide the neuronal fluorescent signals to a CMOS detector. Additionally, a gradient-index (GRIN) lens can be implanted into the animal’s brain to act as a relay (Figure 1).

Compared to the conventional two-photon microscope, the cost of a one-photon miniscope system is much lower, at approximately USD 1500. The only consumables are the relay lens and the baseplate, both of which are reusable. Additionally, miniscopes can provide a large field of view with an impressive fast sampling frequency (>30 Hz). The total system is around 3 g and allows for easy attachment to small animals such as bats, birds, and rodents, enabling unrestricted and more complex behavioural studies (Figure 2). Figure 3 provides an example from our previous work using a miniscope to study the hippocampal calcium activity in mice [38].

One major disadvantage of this system is interference from background noise, as the excitation light can also stimulate neurons in neighbouring brain regions near the focal plane. However, this limitation could be turned into an advantage by applying advanced calcium signal analysis algorithms that enable more precise volumetric signal decoding [39]. As the miniscope system is based on one-photon microscopy, it is more limited in imaging depth due to brain tissue scattering, typically effective up to 200 µm. This is much lower than two-photon microscopy, which allows a maximum imaging depth of around 500–1500 µm. Since the invention of miniscope technology in 2011, numerous open-source projects have emerged, such as the UCLA miniscope, FinchScope, CHEndoscope, Ninscope, and Miniscope3D, along with commercial options like the Inscopix miniscope and Doric Lenses [27,40,41,42]. Among these, the UCLA miniscope is perhaps the most influential, with multiple iterations including the V3 and V4 miniscopes, WireFree miniscope, LFOV miniscope, and the recent miniaturised two-photon microscope [43,44].

## 3. Calcium Signalling in Neurological Disorders

### 3.1. Amyotrophic Lateral Sclerosis (ALS)

ALS is a neurological disorder that selectively impairs the function of motor neurons in both the brain and spinal cord [45,46,47]. While 90–95% of cases are found to be sporadic [48], a small number of cases exhibit familial aggregation linked to specific gene mutations, notably C9orf72, SOD1, FUS, SIGMAR1, and TDP43 [49]. Interestingly, these mutations have also been occasionally identified in sporadic cases [49]. Substantial evidence suggests that disruptions in calcium homeostasis significantly contribute to the pathogenesis of ALS [14,50,51,52]. Previous studies using ALS mouse models with mutations in the TDP-43 or FUS gene have observed impaired physical and functional coupling between the ER and mitochondria [53,54]. This alteration affects mitochondrial calcium uptake and influences various calcium-related signalling pathways, resulting in calcium accumulation in the cytosol. Low mitochondrial calcium levels impair cellular energy production, while calcium accumulation in the cytosol overwhelms the calcium-buffering system and contributes to the excitotoxicity process. On the other hand, calcium overload in the mitochondria has been reported in motor nerve terminals from ALS patients and SOD1 mutant mice [55,56]. Excessive calcium accumulation can lead to mitochondrial permeability transition and promote cell death. The observed contradictions may be related to gene mutations or suggest that ALS involves multiple phases of calcium dysregulation. Calcium-binding proteins (CaBPs) also regulate cellular calcium homeostasis [55,57]. Prior works on SOD1-mutant mice have shown that motor neurons in the spinal and hypoglossal nuclei with low levels of CaBPs are more susceptible to degeneration than oculomotor neurons with high levels of CaBPs [57]. Furthermore, CaBP overexpression has been found to delay the onset of ALS symptoms, while low levels of CaBPs have been observed in the brains of ALS patients [58,59,60]. In addition to CaBPs, several glutamate receptors have been identified as significant regulators to maintain calcium homeostasis in ALS [47,55,61,62]. Blocking calcium-permeable AMPA receptors in motor neurons has shown a protective effect against neuronal damage [63,64], while the activation of NMDA receptors leads to increased calcium influx and results in neuronal excitotoxicity [62].

The miniscope has proven to be a valuable tool for investigating neuronal population-level abnormalities in ALS studies. In a recent study conducted by Liang et al., a customised miniscope was used to compare the calcium activity in pyramidal cells of the prefrontal cortex between TDP-43-depleted and -intact mice [65]. Researchers performed in vivo optical recordings over a period of seven months and found that short-term TDP-43 depletion triggered a cascade of events involving hyperactive calcium activity, rapid activity decline, and eventual neuron loss. In another study, Brailoiu et al. utilised the miniscope to examine how Sigma-1R modulates the permeability of the blood–brain barrier (BBB), which has been observed to be compromised in ALS patients and animal models [66]. Researchers showed that Sigma-1R activation increased BBB permeability of prefrontal cortex neurons in rats, as indicated by elevated levels of sodium fluorescein. Since calcium signalling is vital for maintaining the integrity and function of BBB, the observed results may indicate a disruption in this process.

### 3.2. Alzheimer’s Disease (AD)

AD is a neurological disorder that impairs both the formation and retention of memories [67]. Although most AD cases (>95%) are sporadic, with symptom onset typically occurring in individuals over 65 years of age [67,68], several genetic factors have been identified that increase the risk of developing AD. Among these, APOE variants are the most commonly reported and can triple the likelihood of developing the disease [69,70]. In contrast, familial AD accounts for only 1 to 5% of cases and is attributed to autosomal dominant mutations in genes encoding the ER membrane proteins, namely presenilin 1 and presenilin 2 [11,68,71]. Although the cause of AD is unknown, the dominant hypothesis in the field is the “amyloid cascade hypothesis”. This pathogenic model attributes the development of AD to the abnormal accumulation of amyloid beta (Aβ) peptides in brain regions associated with memory and cognition, such as the hippocampus and related cortices. The aggregation of Aβ peptides possibly stems from the proteolytic cleavage of amyloid precursor protein by alpha and beta secretases, which are associated with mutated presenilins [11,71,72,73]. However, none of the current therapeutic strategies targeting Aβ plaques have been successful in preventing AD pathogenesis despite providing varying levels of plaque removal [74,75].

Another hypothesis that has recently received considerable attention is the “calcium hypothesis”. This hypothesis speculates that Aβ plaques lead to the aberrant regulation of calcium homeostasis and impair the function of the neural network [76]. In patients with AD, the accumulation of Aβ is often associated with abnormal calcium influx and impaired coupling between the ER and mitochondria. Aβ peptides enhance calcium channel permeability and result in elevated cytoplasmic calcium levels, which increase neuronal vulnerability to excitotoxicity and apoptosis [76,77,78]. Although various working mechanisms have been proposed, the majority of studies suggest that excessive calcium release from the ER through the IP3R and RYR calcium channels is a critical factor. The most representative evidence comes from presenilin mutant knock-in mice, where cortical neurons release substantial amounts of calcium ions upon the activation of IP3R [79,80]. NMDAR-mediated calcium imbalance has also been proposed to contribute to AD. Aβ peptides interact with NMDARs and increase calcineurin deposition at synapses, which depresses synaptic activity and inhibits long-term potentiation encoding [68]. Furthermore, aberrant NMDAR-generated signals disrupt the normal calcium-dependent signalling cascade that remodels dendritic spines, thereby compromising synaptic structural plasticity [81]. This deficiency is evident in both AD patients and mouse models, with a significant absence of mature dendritic spines [82,83,84]. In addition, an FDA-approved NMDAR antagonist, memantine, has been shown to temporarily improve cognitive function in AD patients [85].

The miniscope has recently gained widespread use in AD research. Zhang et al. measured and compared hippocampal neuronal calcium activity in the 5xFAD AD mouse model and wild-type mice using a miniscope [86]. Their study demonstrated that by the age of 4 months, 5xFAD mice exhibited amyloid plaque accumulation, reduced neuronal calcium activity during immobile states, and unreliable spatial tuning of hippocampal neurons to environmental locations. However, their object location memory remained comparable to that of wild-type mice at this stage. By 8 months, 5xFAD mice displayed object location memory deficits linked to progressive degradation of spatial encoding. These results demonstrate a strong connection between early hippocampal neural network dysfunction and the development of memory deficits later in disease progression, suggesting it to be a potential biomarker for earlier AD diagnosis. In another study using the miniscope, Lin et al. examined hippocampal calcium activity in a 3xTg AD mouse model, which was characterised by amyloid plaques, neurofibrillary tangles, and age-related behavioural deficits [87]. The researchers evaluated the neural encoding of the environmental location in mice during free exploration and route-running tasks. The 3xTg AD mice exhibited significantly higher calcium activity rates as well as lower spatial sensitivity and specificity in CA1 cells compared to controls, particularly in older mice. Additionally, the locomotor speed had a more substantial impact on calcium activity amplitude in these mice compared to controls. These results suggest that spatial encoding defects in hippocampal neural circuits are linked to AD-related memory and behavioural deficits. Similar spatial coding deficits were also observed by Murano et al. in the dentate gyrus neurons of αCaMKII heterozygous knockout mice [88]. In addition to spatial coding, criticality may serve as another potential biomarker for AD. Criticality in the brain describes a condition of dynamic equilibrium between order and chaos in neural activity, typically linked to optimal performance [89]. In a recent study from our lab, we explored hippocampal neuronal calcium activity in a psychopharmacological mouse model of Alzheimer’s disease (AD) using scopolamine during a novel object recognition task [90]. After the administration of scopolamine, the dynamics of the hippocampal neural network significantly deviated from criticality during the task, while the saline control group did not exhibit this deviation. These findings possibly explain our previous observation that scopolamine administration changes the properties of the hippocampal neural network in a linear track experiment [38]. Aβ has also been found to impact neuronal calcium activity in a state-dependent manner. Zhou et al. investigated the hippocampal activity in an APP/PS1 AD mouse model and found that Aβ induced neuronal hyperactivity during exploratory behaviour and slow-wave sleep but suppressed the activity during quiet wakefulness and rapid eye movement sleep [91]. These findings highlight the importance of developing therapies that specifically target the state-dependent effects of Aβ on brain functions in AD. Besides impaired episodic memory, AD is also characterised by profound deficits in visual cognition [92]. Using the Inscopix miniscope, Parka et al. investigated visual stimuli-evoked neuronal calcium activity in the primary visual cortex of the rTg4510 tauopathy mouse model during the early stages of the disease, prior to the onset of neurodegeneration [93]. In rTg4510 mice, neurons in the visual cortex exhibited reduced responsivity and were insensitive to GABAergic modulation. Additionally, input and output synchronicity was found to be significantly impaired in the basal state. These findings indicate that impairments in visual cortical processing may serve as a biomarker in the early stages of neurodegenerative tauopathies. Furthermore, these results may also help explain the recent findings that rhythmic light flicker alleviates symptoms in AD patients and reconstructs hippocampal neural networks [94,95]. In addition to rhythmic light flicker, deep brain stimulation is another promising therapeutic method for AD. Shoob et al. found that deep brain stimulation in the thalamic nucleus reuniens restored anaesthesia-induced suppression of CA1 calcium activity in APP/PS1 mice and mitigated age-dependent memory decline [96].

### 3.3. Parkinson’s Disease (PD)

PD is a complex progressive neurological disorder [97]. The clinical diagnosis of PD primarily depends on the manifestation of bradykinesia, a cardinal motor feature characterised by slowness of movement, along with at least one additional cardinal sign such as rigidity or resting tremor [98]. At the cellular level, PD is characterised by the progressive loss of dopaminergic neurons in the substantia nigra pars compacta (SNc) and the abnormal accumulation of α-synuclein in the form of Lewy bodies in the affected brain regions [99,100]. PD primarily occurs sporadically without a clear genetic cause, although a small proportion (5–10%) of cases has been associated with genetic mutations at various loci known as PARKs [99]. Notably, the SNCA gene, classified under PARK1, is responsible for encoding α-synuclein. Several lines of evidence suggest that PARK genes encode the proteins essential for mitochondrial function [101]. The disruption of these proteins leads to impaired mitochondrial activity and a disturbance in cellular calcium homeostasis. Additionally, impaired calcium communication between the mitochondria and ER has also been reported [101]. Similar to the findings in ALS, contradictory effects on mitochondrial calcium uptake have been observed in PD, which can be attributed to either enhanced or reduced ER calcium release [102,103]. This variation may result from differences in cell models or the diverse expression levels attained in various experiments, but the altered ER–mitochondrial calcium coupling that occurs during the early stages of PD is a consensus in the field [51].

In addition to mitochondrial dysfunction, alterations in voltage-gated L-type calcium channels are associated with PD, especially the CaV1.3 subtype. SNc dopaminergic neurons play a critical role in regulating dopamine levels within the brain through their unique pace-making activity. This intrinsic activity is partially driven by CaV1.3 channels and is altered in PD. Studies conducted on brain tissues from PD patients have shown an increased expression of the neuronal CaV1.3 subtype during the early stages of the disease, even before the appearance of pathological changes [104]. The overexpression of CaV1.3 channels in SNc neurons results in elevated calcium influx, causing disruptions in the normal pace-making activity and creating excessive metabolic stress. This metabolic burden, combined with the disrupted calcium homeostasis, renders SNc neurons more vulnerable to degeneration. As evidence for this hypothesis, the pharmacological blockade of CaV1.3 channels using isradipine has been found to restore pace-making activity and demonstrate protective effects on SNc neurons in rodent models of PD [105,106]. Chronic administration of isradipine not only reduces cytosolic calcium levels in SNc neurons but also alleviates mitochondrial oxidative stress [107]. Additionally, the knockdown of CaV1.3 channels leads to decreased dendritic calcium oscillations in SNc neurons [107].

Both the ventral tegmental area (VTA) and the SNc are part of the mesolimbic and nigrostriatal dopaminergic systems. However, the VTA is not as directly affected as the SNc in PD [108]. In comparison to dopamine neurons located in the VTA, SNc dopaminergic neurons display lower expression levels of the calcium-buffering protein (CaBP) [108]. In an animal model of PD, the overexpression of CaBP-28 k has been shown to promote neurite outgrowth in dopaminergic neurons and provide some protection against degenerative processes [109]. These results suggest that CaBP overexpression could be a potential therapeutic approach for PD.

The miniscope enables the real-time imaging of calcium dynamics in freely behaving animals, allowing the observation of changes at the neural circuits level in PD. Parker et al. investigated the calcium activity of striatal spiny projection neurons (SPNs) in a 6-OHDA-induced PD mouse model using the Inscopix miniscope [110]. Their results demonstrated that dopamine depletion in PD mice led to hypoactivity in SPNs of the direct pathway and hyperactivity in SPNs of the indirect pathway. Additionally, treatment with the dopamine precursor L-DOPA and dopamine D2 receptor agonists reversed these abnormalities. L-DOPA is an FDA-approved medication for the treatment of PD that can help increase dopamine levels in the brain. However, patients may develop a movement disorder known as dyskinesia with long-term use. In the experiment, the opposite pathophysiology was observed in L-DOPA-induced dyskinesia mice. These findings indicate that future therapies should address the concurrent biological changes that lead to disruptions in striatal neural ensemble coding. In another study, Trevathan et al. used the same 6-OHDA PD mouse model to investigate the therapeutic effects of deep brain stimulation in the subthalamic nucleus [111]. Utilising the miniscope, researchers analysed changes in the neuronal calcium activity within the striatum during the stimulation. The findings showed that stimulation of the subthalamic nucleus elicited distinct calcium responses depending on physiological conditions. Significant differences were observed between anaesthetised and awake states, as well as during rest and movement in the awake state. These preliminary data highlight the importance of exploring the mechanisms of deep brain stimulation in awake and behaving pathological models.

### 3.4. Huntington’s Disease (HD)

HD is an autosomal dominant genetic disorder caused by a CAG repeat expansion in the HTT gene [112]. This disease has a major effect on a patient’s mood and cognitive and motor functions, with symptoms typically emerging in early adulthood. Initial signs of HD may include depression and involuntary movements [113]. Medium spiny GABAergic inhibitory neurons in the striatum are most affected in HD, although pyramidal neurons in the cerebral cortex are also impacted [111,112,113]. Disrupted calcium homeostasis is considered a key contributor to HD pathology, since it is commonly observed prior to striatal dysfunction and the onset of HD symptoms [114,115,116]. One of the primary causes of disrupted calcium homeostasis appears to be the dysfunction of certain calcium release channels [112,113]. The mutant HTT (mHTT) protein binds directly and specifically to the C-terminal region of the IP3R1 receptors on the ER membrane surface. The alteration increases the receptor’s sensitivity to IP3 activation, which causes calcium ions to be released from the ER into the cytoplasm. This release promotes mitochondrial calcium uptake and impairs neuronal functions. However, viral delivery of the IC10 peptide in a YAC128 HD transgenic mouse model disrupts the interaction between mutant huntingtin (mHTT) and IP3R1, protecting neurons both in vitro and in vivo [117]. The glutamate-gated calcium channels NMDARs have also been found to impair calcium homeostasis in HD [118]. The expression of mHTT promotes extrasynaptic NMDAR expression and leads to enhanced NMDAR activation [119]. This heightened activation increases calcium influx into neurons, ultimately resulting in neuronal cell death. Additionally, both in vivo and in vitro studies have demonstrated that the pharmacological blockade of NMDARs helps attenuate excessive calcium influx, providing protection against excitotoxic damage [120,121]. Beyond that, the loss of calcium binding proteins is another associated phenomenon with the development of HD. Neuropathological studies have identified a selective loss of CB-D28k-expressing neurons in the striatum of HD patients, which reduces the neuronal calcium-buffering capacity [122,123]. However, it remains unclear whether this is a causal factor in HD or just an associated phenomenon.

The application of miniscope technology in HD research has been relatively limited so far. In a proof-of-concept study, Barry et al. utilised the UCLA V3 miniscope to record the calcium activity in neurons from the cortical layers V to VI in symptomatic R6/2 mice, a transgenic model expressing a fragment of mHTT [124]. The researchers observed a reduction in the amplitude of calcium transients, consistent with previous observations made using two-photon laser scanning microscopy [125]. In a separate experiment, the same research group labelled specific interneuron populations and observed their calcium activity in the D1-Cre HD mouse model. Although the experiment is still in the preliminary stage, the results highlight the potential for using the miniscope to study the development and alterations of striatal calcium circuits in HD.

### 3.5. Schizophrenia

Schizophrenia is a chronic psychiatric disorder characterised by dysfunctions across multiple neural circuits and signalling pathways [126]. Its development is complex, involving variations in over 1800 genes and polymorphisms that impair normal neurotransmitter function [127]. Notably, many of these genetic mutations have been reported to impair cellular calcium signalling and homeostasis [128]. The most frequently reported genes include those that directly encode L-type voltage-gated calcium channels, such as CACNA1C and CACNB2; genes involved in glutamatergic and calcium signalling, specifically GRM3; and genes that indirectly affect intracellular calcium levels, such as RGS4 and GAP-43 [129,130,131,132]. Additionally, a recent clinical study reported that dihydropyridine treatment significantly reduced the risk of psychiatric rehospitalization in schizophrenia patients. This class of compounds selectively binds to L-type voltage-gated calcium channels and can easily permeate the blood–brain barrier [133].

An additional factor that disrupts intracellular calcium homeostasis in individuals with schizophrenia is associated with calcium-buffering proteins (CBPs) like parvalbumin, calbindin, and calretinin [134]. The abnormal expression of CBPs and disrupted calcium homeostasis in schizophrenia have been reviewed by Eyles et al. and are briefly summarised here [134]. Parvalbumin is predominantly expressed in a specific subset of GABAergic interneurons. In contrast, calbindin and calretinin are widely expressed, but their levels are significantly lower in pyramidal cells than in interneurons within the neocortex and hippocampus. There is a growing number of pathological reports documenting subtle alterations in interneurons containing these CBPs in patients with schizophrenia. These proteins have been shown to provide survival benefits to neurons and enhance their capacity to maintain firing in some studies. However, the literature on the functions and effects of these proteins is filled with studies that have small sample sizes and often yield contradictory results. Nevertheless, most studies have observed decreased levels of CBPs in interneurons, which disrupt calcium homeostasis and lead to pro-apoptotic responses.

Schizophrenia affects not only the calcium dynamics of individual neurons but also the calcium communication among neuronal populations. Liang et al. examined the neuronal population activity in the medial prefrontal cortex (mPFC) of mice during real-time social exploration using the miniscope [135]. They investigated the mPFC calcium activity both with and without the administration of phencyclidine (PCP), a psychedelic drug known to induce schizophrenia-like symptoms in rodents. In normal mice, the researchers identified distinct neuronal ensembles that were either “ON” or “OFF”, exhibiting opposing network activities to encode real-time behavioural information about the novelty and significance of social targets. However, after the administration of PCP, the animals exhibited inappropriate social exploratory behaviour, which was associated with dysfunctions in these ensembles. In a similar study using the miniscope, Grieco et al. found that subanaesthetic ketamine increased the calcium activity of mPFC excitatory neurons in vivo [136]. Similar to PCP, which acts as an NMDA receptor antagonist, ketamine can also induce symptoms resembling those of schizophrenia. After ketamine treatment, the firing frequency of calcium events in mPFC excitatory neurons increased significantly. Moreover, the peak amplitudes and overall intensity of these calcium events remained elevated even after 24 h, being potentially influenced by neuregulin-1 signalling mediated by parvalbumin interneurons. In addition to mPFC, Masuda et al. investigated the effects of ketamine on neuronal calcium activity within the hippocampal neural network [137]. Ketamine rapidly elevated neuronal calcium event rates and disrupted the temporal firing-rate coupling between neurons, resulting in a sustained reorganization of spatial representations. These findings illustrate a novel mechanism of ketamine’s effects based on neural plasticity.

## 4. Conclusions

Calcium signalling and calcium homeostasis are essential for proper neuronal circuit function. The dysregulation of calcium signalling pathways along with disrupted calcium homeostasis is a characteristic feature of NDs. Miniscope technology provides a novel approach for studying calcium activity of large neuronal ensembles in freely moving animals, with the advantage of achieving cellular level resolution. Although the use of miniscope technology for investigating NDs has seen a considerable increase in recent years (Table 1), it has yet to reach the same level of widespread adoption as the two-photon microscope. This review aims to highlight the promising potential of miniscope technology for researchers studying NDs. Additionally, the miniscope can be adapted for use as an optical brain–computer interface, which may enhance communication for individuals affected by NDs [138].

Most previous NDs studies focus on the calcium properties of individual neurons. However, the results from miniscope studies indicate that the neurological deficits observed in NDs may stem from calcium functional impairments within the neural network rather than being solely attributed to individual neurons. These findings suggest that therapies targeting the reconstruction and restoration of neural circuit function might be more effective than those focusing solely on the attributes of individual neurons.

## Figures and Tables

**Figure 1 cells-14-00004-f001:**
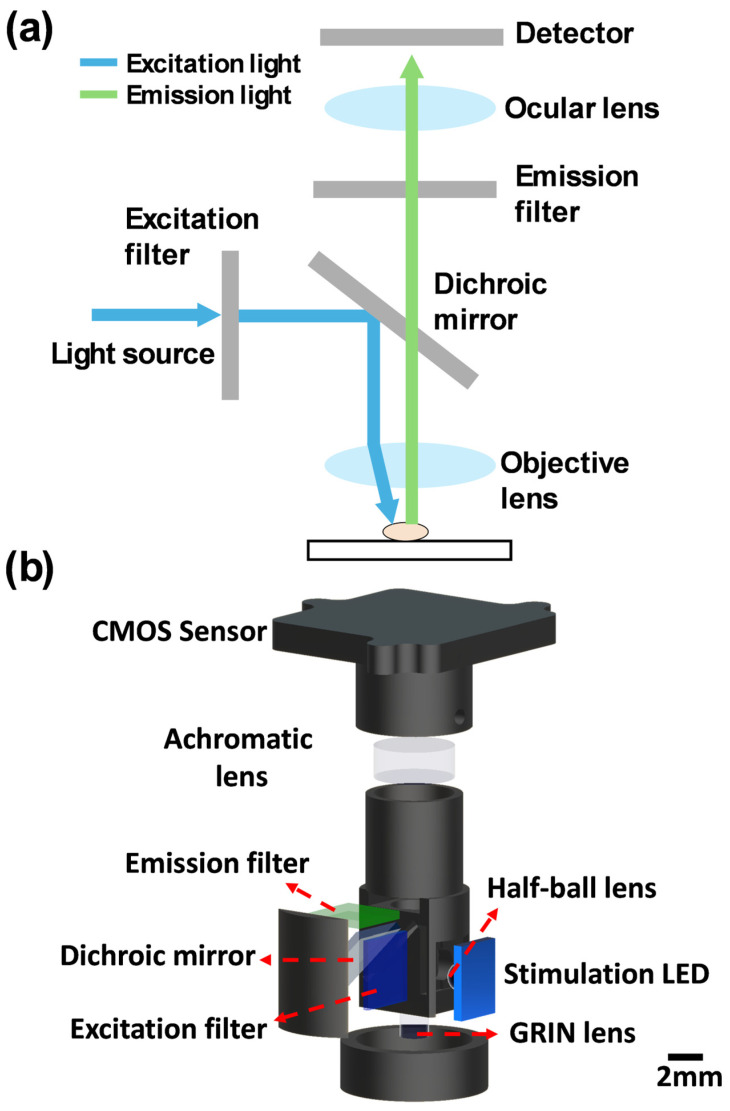
A typical structure of a conventional fluorescence microscope and the miniscope. (**a**) Conventional fluorescence microscope. (**b**) Miniscope: An LED emits light at a wavelength that excites the fluorophore, which is collimated by a half-ball lens. The excitation light passes through an excitation filter to reduce background noise before being reflected by a dichroic mirror. A GRIN lens focuses the light to activate fluorophores in neurons and collects the fluorescent signals. These signals are then directed back through the dichroic mirror and an emission filter, isolating the desired wavelength. Then, an achromatic lens focuses the signals onto a CMOS sensor.

**Figure 2 cells-14-00004-f002:**
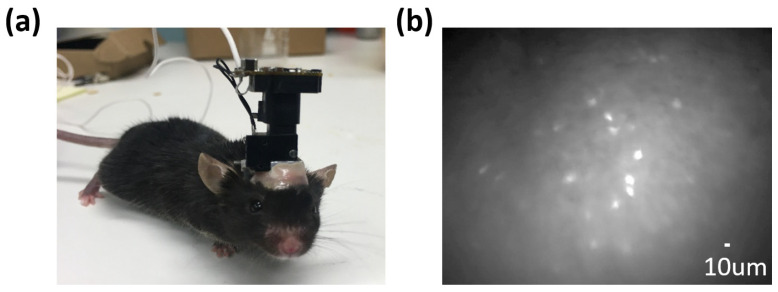
The miniscope facilitates the recording of neuronal calcium signals in freely moving mice. (**a**) An example of a C57BL/6 mouse wearing the UCLA miniscope (version 3). (**b**) A representative image showing hippocampal CA1 neurons captured using the miniscope. Neurons are labelled with GCamp6f. Typically, over 100 neurons can be observed within the field of view.

**Figure 3 cells-14-00004-f003:**
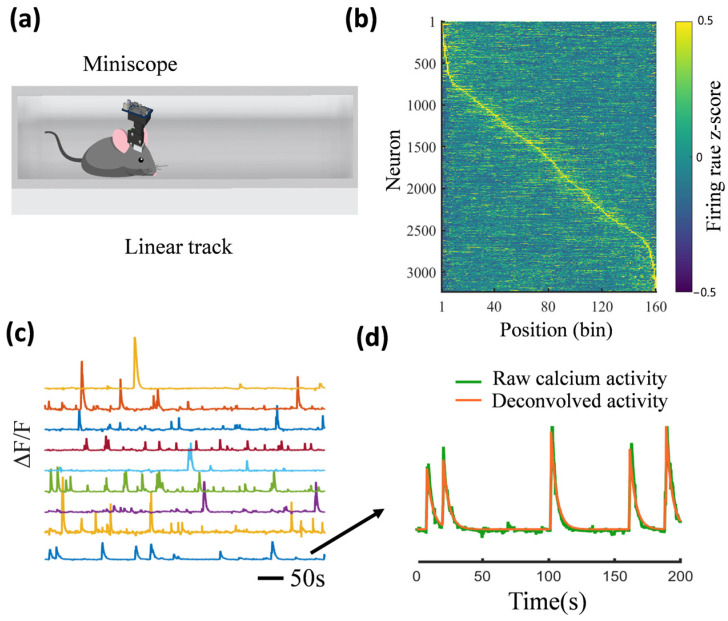
An example of using the miniscope to study the calcium activity of hippocampal neurons in mice. (**a**) A mouse wearing a miniscope traverses a linear track. (**b**) An example showing that the calcium activity of hippocampal neurons exhibits spatial sensitivity when mice traverse a linear track. (**c**) An example of raw fluorescent intensity in the detected neurons captured by the miniscope. (**d**) An example of a neuron’s deconvolved calcium activity.

**Table 1 cells-14-00004-t001:** Studies of neurological disorders using a miniscope.

Neurological Disorders	Miniscope Device	Year	Authors
ALS	Customised	2022	Liang et al. [65]
ALS	Doric Lenses	2024	Brailoiu et al. [66]
AD	UCLA	2023	Zhang et al. [86]
AD	UCLA	2022	Lin et al. [87]
AD	Inscopix	2022	Murano et al. [88]
AD	UCLA	2023	Habibollahi et al. [90]
AD	UCLA	2023	Sun et al. [38]
AD	Inscopix	2022	Zhou et al. [91]
AD	Inscopix	2023	Parka et al. [93]
AD	Inscopix	2023	Shoob et al. [96]
PD	Inscopix	2018	Parker et al. [110]
PD	Inscopix	2021	Trevathan et al. [111]
HD	UCLA	2022	Barry et al. [124]
Schizophrenia	Customised	2018	Liang et al. [135]
Schizophrenia	UCLA	2021	Grieco et al. [136]
Schizophrenia	UCLA	2023	Masuda et al. [137]

## Data Availability

No new data were created or analysed in this study.

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
