# Peer review of "Calcium Signalling in Neurological Disorders, with Insights from Miniature Fluorescence Microscopy"

_cells, 2024, doi:10.3390/cells14010004_

Round 1
Reviewer 1 Report
Comments and Suggestions for Authors
This is a timely review on the posibilities of the use of the miniscope to understand contributions of intracellular calcium dyshomeostasis to neurological diseases. The minireview is very well structured, provide useful information and may update quickly the reader on this important issue in calcium signaling. Some minor changes are suggested for improvement.
In the last paragraph of first page (introduction section) it is stated that changes in calcium signaling can disrupt calcium homeostasis. I consider this sentence wrong as changes in calcium homeostasis is which leads to changes in calcium signaling.
In second page (Introduction section), it is stated that cell surface receptors can activate receptor-operated channels. This is confusing. It is simpler to stated that calcium entry can be induced ligand-gated calcium channels as, for instance, NMDA receptors.
Please add in vivo after ....for studyng calcium dynamics in ND.
Genetically encoded calcium indicators are good not only because they can be targeted to specific cell populations but also to specific cell organelles with the proper calcium affinity. Please include it in point 2 of introduction.
page 5. ....while calcium accumulation in the cytosol (not cytosolic)
As a suggestion, I will increase the siz of Figure 1 to see better the parts of the Miniscope, even in parallel with a regular fluorescence microscope.
A final suggestion is to include the calcium probe used in the description of miniscope studies and method of cell targeting if any. This may help the reader.
Author Response
This is a timely review on the possibilities of the use of the miniscope to understand contributions of intracellular calcium dyshomeostasis to neurological diseases. The minireview is very well structured, provide useful information and may update quickly the reader on this important issue in calcium signaling. Some minor changes are suggested for improvement.
(1) In the last paragraph of first page (introduction section) it is stated that changes in calcium signaling can disrupt calcium homeostasis. I consider this sentence wrong as changes in calcium homeostasis is which leads to changes in calcium signaling.
We appreciate the valuable comments from the reviewer. We agree calcium homeostasis typically leads to changes in calcium signalling. On the other hand, calcium signalling can also influence calcium homeostasis. However, disturbances in calcium homeostasis generally have a more direct impact on signalling. We have changed this statement to “Dysfunction in calcium signalling can lead to either excessive or insufficient calcium levels within neurons.”
(2) In second page (Introduction section), it is stated that cell surface receptors can activate receptor-operated channels. This is confusing. It is simpler to stated that calcium entry can be induced ligand-gated calcium channels as, for instance, NMDA receptors.
We agree this is confusing. We have modified the manuscript as per the comment.
(3) Please add in vivo after ....for studying calcium dynamics in ND.
We have emphasized in vivo in the manuscript.
(4) Genetically encoded calcium indicators are good not only because they can be targeted to specific cell populations but also to specific cell organelles with the proper calcium affinity. Please include it in point 2 of introduction.
We have included this in point 2 of introduction.
(5) page 5. ....while calcium accumulation in the cytosol (not cytosolic)
Sorry for the typo. We have replaced “cytosolic” with “cytosol”.
(6) As a suggestion, I will increase the size of Figure 1 to see better the parts of the Miniscope, even in parallel with a regular fluorescence microscope.
We have increased the figure size and added a regular fluorescence microscope in the figure.
(7) A final suggestion is to include the calcium probe used in the description of miniscope studies and method of cell targeting if any. This may help the reader.
Calcium probes, such as calcium ion-selective microelectrodes or calcium-selective field-effect transistors, utilize calcium-selective membranes that specifically bind to calcium ions to monitor calcium ion concentrations and activities [33–34]. When exposed to a solution containing calcium ions, this selective binding generates a charge imbalance across the membrane, resulting in an electrical potential. Compared to calcium imaging methods, these techniques offer superior temporal resolution. However, they are not ideal for recording large neuronal populations and are primarily used in in vitro experiments due to the challenges associated with miniaturizing the recording device. Additionally, although brain tissue can maintain a relatively stable pH and temperature, which does not severely affect recording accuracy, interference from other ions and membrane degradation over time may reduce its sensitivity and cause signal drift, leading to inaccurate measurements. One major limitation of these techniques is their non-specificity. Calcium probes do not possess intrinsic capabilities for targeting specific neurons based on molecular markers. As a result, researchers often rely on electrophysiological signatures (e.g., firing frequency or firing patterns) to identify and differentiate between neuronal subtypes during experiments. We have mentioned these in the manuscript.
Reviewer 2 Report
Comments and Suggestions for Authors
In this review, the authors have made a general overview of the role of Calcium signalling alterations in several neurological diseases and the potential therapeutical strategies to apply, also exploring the use of a relatively recent method, which they are experts in, to investigate the phenomenon. They have been very detailed, (they have cited a huge amount of differentiated and pertinent papers) examining the different hypotheses and also talking about controversial data. I appreciated the division into paragraphs by neurological disorder addressed and the related use of the miniscope.
So in my opinion, the review can be accepted for publication in Cells, given the general interest in the topic and how carefully the authors have addressed it. Furthermore, the topic fits perfectly within the themes of the Special Issue. I suggest below only a few minor points to address:
- the review described the many advantages of using the miniscope, compared to the more widely used 2-photon microscope. Still, I was wondering if its use had any disadvantages (e.g., high costs, conflicting data?), and if so, they should be discussed.
- in paragraph 3.1 it would be better to replace "calcium accumulation in the cytosolic" with something like "cytoplasm" or “cytosolic compartment”.
Author Response
In this review, the authors have made a general overview of the role of Calcium signalling alterations in several neurological diseases and the potential therapeutical strategies to apply, also exploring the use of a relatively recent method, which they are experts in, to investigate the phenomenon. They have been very detailed, (they have cited a huge amount of differentiated and pertinent papers) examining the different hypotheses and also talking about controversial data. I appreciated the division into paragraphs by neurological disorder addressed and the related use of the miniscope. So in my opinion, the review can be accepted for publication in Cells, given the general interest in the topic and how carefully the authors have addressed it. Furthermore, the topic fits perfectly within the themes of the Special Issue. I suggest below only a few minor points to address:
(1) the review described the many advantages of using the miniscope, compared to the more widely used 2-photon microscope. Still, I was wondering if its use had any disadvantages (e.g., high costs, conflicting data?), and if so, they should be discussed.
We would like to thank the reviewer for providing us with valuable feedback. Compared to the conventional two-photon microscope, the cost of a one-photon miniscope system is much lower, at approximately $ 1500. The only consumables are the relay lens and the baseplate, both of which are reusable. As the miniscope system is based on one-photon microscopy, it is more limited in imaging depth due to brain tissue scattering, typically effective up to 200 µm. This is much lower than two-photon microscopy, which allows a maximum imaging depth of around 500-1500 µm. We have mentioned these disadvantages in section 2.
(2) in paragraph 3.1 it would be better to replace "calcium accumulation in the cytosolic" with something like "cytoplasm" or “cytosolic compartment”.
Sorry for the typo. We have replaced “cytosolic” with “cytosol” in paragraph 3.1.
Reviewer 3 Report
Comments and Suggestions for Authors
The review article by Sun et al. focus on the critical role of calcium signaling in several widespread neurological diseases (NDs), such as Alzheimer's disease (AD), Parkinson's disease (PD), Huntington's disease (HD), and schizophrenia, which are some of the most challenging conditions worldwide. The introduction provides a brief summary of the significance of calcium signaling in NDs, which overtime can trigger apoptosis and contribute to development of NDs, followed by an explanation of Miniature one-photon microscopy. The history and principles behind this innovative technique are well described; with figure 1 accompanying its look and figure 2 presents how other researchers have utilized this compact microscope to monitor calcium activity in live brain imaging and behaving animals, are presented clearly and provide valuable insights for a wide range of researchers. The paper is easy to follow, and in subsequent sections, the authors explore the role of calcium signaling in various neurological disorders as they described in introduction part. The conclusions are consistent, and the references are accurate. The paper also discusses the potential use of miniscope technology in research and its therapeutic implications and its limitation. Overall, the article is well written, clear, and both informative and understandable.
Author Response
The review article by Sun et al. focus on the critical role of calcium signaling in several widespread neurological diseases (NDs), such as Alzheimer's disease (AD), Parkinson's disease (PD), Huntington's disease (HD), and schizophrenia, which are some of the most challenging conditions worldwide. The introduction provides a brief summary of the significance of calcium signaling in NDs, which overtime can trigger apoptosis and contribute to development of NDs, followed by an explanation of Miniature one-photon microscopy. The history and principles behind this innovative technique are well described; with figure 1 accompanying its look and figure 2 presents how other researchers have utilized this compact microscope to monitor calcium activity in live brain imaging and behaving animals, are presented clearly and provide valuable insights for a wide range of researchers. The paper is easy to follow, and in subsequent sections, the authors explore the role of calcium signaling in various neurological disorders as they described in introduction part. The conclusions are consistent, and the references are accurate. The paper also discusses the potential use of miniscope technology in research and its therapeutic implications and its limitation. Overall, the article is well written, clear, and both informative and understandable.
We sincerely thank the reviewer for the time and feedback on our paper.